# Pathological Responses of Cardiac Mitochondria to Burn Trauma

**DOI:** 10.3390/ijms21186655

**Published:** 2020-09-11

**Authors:** Meijing Wang, Susan R. Scott, Leonidas G. Koniaris, Teresa A. Zimmers

**Affiliations:** 1Department of Surgery, Indiana University School of Medicine, Indianapolis, IN 46202, USA; surscott@iu.edu (S.R.S.); lkoniari@iu.edu (L.G.K.); zimmerst@iu.edu (T.A.Z.); 2Simon Cancer Center, Indiana University, Indianapolis, IN 46202, USA; 3Indiana Center for Musculoskeletal Health, Indianopolis, IN 46202, USA; 4Center for Cachexia Research Innovation and Therapy, Indiana University Purdue University Indianapolis, Indianapolis, IN 46202, USA

**Keywords:** burn trauma, mitochondrial damage, cardiac dysfunction, aging, gender

## Abstract

Despite advances in treatment and care, burn trauma remains the fourth most common type of traumatic injury. Burn-induced cardiac failure is a key factor for patient mortality, especially during the initial post-burn period (the first 24 to 48 h). Mitochondria, among the most important subcellular organelles in cardiomyocytes, are a central player in determining the severity of myocardial damage. Defects in mitochondrial function and structure are involved in pathogenesis of numerous myocardial injuries and cardiovascular diseases. In this article, we comprehensively review the current findings on cardiac mitochondrial pathological changes and summarize burn-impaired mitochondrial respiration capacity and energy supply, induced mitochondrial oxidative stress, and increased cell death. The molecular mechanisms underlying these alterations are discussed, along with the possible influence of other biological variables. We hope this review will provide useful information to explore potential therapeutic approaches that target mitochondria for cardiac protection following burn injury.

## 1. Introduction

Despite advances in treatment and care, burn trauma remains the fourth most common type of traumatic injury. Each year burns affect around 300 million patients worldwide and result in nearly 4000 deaths in the United States [1]. According to the American Burn Association, the cause of burn injury death is often burn complications, including hypovolemic shock, multi-organ failure, respiratory problems, or infection (sepsis). Of note, impaired cardiovascular function has been noticed as one of the main determinants of acute phase responses of multiple organ dysfunction following severe thermal injury [2]. After burn trauma, damaged myocardial structure, increased myocyte death, and depressed cardiac function are observed [3,4]. If more severe, depression in heart contractility and possibly both left and right heart failure can happen [3]. Burn-induced cardiac failure is a key factor for patient mortality, especially during the initial postburn period (the first 24 to 48 h) [5], as autopsy results have shown many pediatric patients present with right heart failure [6]. In addition, post-burn patients may immediately have low cardiac values, characteristic of early shock [7]. Clinical studies have demonstrated that poorer outcomes of burn injury are associated with severe cardiac dysfunction [8,9,10]. Systolic dysfunction (ejection fraction <50%) has been observed in 62% of pediatric burn patients, which is associated with significantly prolonged length of hospital stay [11].

Following burn trauma, cardiac dysfunction is described by slowed isovolumic relaxation, impaired contractility, and decreased diastolic compliance of the left ventricle in animal models [12,13]. The blood flow returned to the heart is reduced due to increased capillary permeability and peripheral constriction following thermal injury [14,15]. This hemodynamic change is believed to be a primary factor resulting in decreased cardiac output at the early stage of burn injury. On the other hand, myocardial contractile depression has been observed in rat hearts treated with serum from burned rats [16], suggesting that burn injury induces the production of myocardial depressant factors in systemic circulation. Importantly, the cardiac derangements can culminate in a state of shock based on animal studies [17,18]. Furthermore, substantial hemodynamic and cardiac dysfunction following major burn injury contribute to the development of sepsis, multiple organ failure, and death [3].

Mitochondria, among the most important subcellular organelles, primarily function as the high-energy molecule adenosine triphosphate (ATP) factory in eukaryotic cells. In addition, mitochondria have been shown to critically impact redox homeostasis, calcium storage and signaling, cell death pathways, and metabolic processes in a variety of cells including cardiomyocytes. In fact, mitochondria are a central player in determining the severity of myocardial damage. Defects in mitochondrial function and structure are involved in the pathogenesis of numerous myocardial injuries and cardiovascular diseases [19]. Therefore, preserving mitochondrial integrity and activity is critical to reducing myocardial impairment and improving cardiac function during severe thermal injury. Given a growing body of studies on burn trauma-induced pathological alterations of mitochondria in diverse tissue/organs, we systematically review the current findings on cardiac mitochondrial pathological changes (Figure 1). We summarize burn-impaired mitochondrial respiration capacity and energy supply, induced mitochondrial oxidative stress, and increased cell death in this article, mainly based on experimental studies. The molecular mechanisms underlying these alterations are discussed, along with the possible influence of other biological variables. We hope this review will provide useful information to explore potential therapeutic approaches that target mitochondria for cardiac protection following burn injury.

## 2. Burn Trauma-Damaged Cardiac Mitochondria

Mitochondrial dysfunction is one of the major driving forces in myocardial damage after injury. Mitochondria are exceptionally abundant in cardiomyocytes (even more numerous than in skeletal muscle) comprising about 35% of myocyte volume in the heart [19] and providing continuous energy supply in supporting cardiac physiological function. Severe burn trauma initiates a profound hypermetabolic response, leading to elevated resting energy expenditures and multi-organ dysfunction throughout the body. Of note, increased mitochondrial uncoupling contributes to hypermetabolic stress response [20], which in turn, results in the higher demand for mitochondrial performance. Maintenance of mitochondrial activity is thus essential to preserve organ function following severe thermal injury.

### 2.1. Burn-Impaired Cardiac Mitochondrial Respiratory Activity and ATP Synthesis

Normal mitochondrial structure and number are important for preserving mitochondrial function. To date, few studies have been conducted to investigate burn-induced alterations of mitochondrial morphology in the heart. Dr. Wen and his colleagues showed a disruption of mitochondrial distribution and polymorphism of the mitochondrial ultrastructure with decreased mitochondrial number and size in rat hearts 24 h after burn [21]. The impaired mitochondrial morphology and structure led to reduced cardiac mitochondrial replication, reduced activity in the transport chains, and ATP deficiencies in cardiac tissue, as a response to burn [21]. Similarly, decreased mitochondrial abundance has been observed in swine heart tissue, associated with reduced ATP-producing efficiency, after severe burn injury [22].

Following burns, functional changes of the NADH respiratory chain have been observed in rat heart mitochondria at an early point of 105 min post-burn [23]. The activities of two respiratory chains (NADH-cytochrome *c* reductase and cytochrome oxidase) have been noticed to decrease at 2 h in burn-injured rat myocardium. Declined activities of succinate-coenzyme Q reductase, succinate-cytochrome *c* reductase, and NADH-coenzyme Q reductase have been found in the rat heart 4 h after burn trauma. These alterations in the electron transport chains were associated with depression of myocardial contractile function [24]. Additional evidence has also suggested decreases in mitochondrial phosphorylate activity and respiratory capacity 8 h after severe thermal injury in rat hearts [25].

Mitochondrial dysfunction in energy supply is attributable to altered signaling pathways that regulate ATP synthesis. It has been reported that severe thermal injury reduced oxygen consumption and the mitochondrial electron transport activity of complex I, III, IV, and V, correlating with decreases in mitochondrial ATP production. The damaged mitochondrial function in rat hearts was mediated through the phosphodiesterase 5 (PDE5)-cyclic GMP protein kinase G (cGMP-PKG) pathway, while blockade of this signaling by sildenafil treatment reversed burn-induced mitochondrial damage [21]. Severe burn injury significantly increased PDE5A expression (both protein and mRNA), decreased cGMP levels, and reduced PKG protein level and activity in rat hearts [26]. Treatment of sildenafil, an inhibitor of PDE5, normalized myocardial levels of PDE5A, cGMP, and PKG, as well as PKG activity after a burn [26]. Indeed, using sildenafil to increase cGMP levels improved cardiac mitochondrial morphology, mitochondrial DNA (mtDNA) replication, mtDNA-encoded gene expressions, and mitochondrial electron transport chain function, as well as mitochondrial complex activity in animal experiments [21]. Furthermore, disrupted mitochondrial respiration capacity has also been observed in the AC16 human cardiomyocyte cell line treated with burn rat serum. An inhibitor of the adenosine monophosphate-activated protein kinase (AMPK), one of the major metabolic energy sensors, exacerbated the burn rat serum-damaged mitochondrial bioenergetics profile, whereas the use of an AMPK activator completely recovered this disrupted respiratory function to levels seen in the untreated serum group [27].

Collectively, these findings suggest that impaired mitochondria are unable to meet the need of cardiac energy after burn traumas, resulting in increased myocyte damage and decreased cardiac function, thus adversely impacting patient outcomes.

### 2.2. Burn-Induced Oxidative Stress

In addition to disrupted mitochondrial respiratory function and impaired energy metabolism, mitochondria play key roles in regulating reactive oxygen species (ROS) defense mechanisms: balancing ROS generation and ROS scavenging in various pathological conditions. It has been shown that burn injury did not affect the activity and expression of cytochrome-c oxidase in cardiac mitochondrial fractions, but led to mitochondrial loss of ROS defense, resulting in post-burn myocardial dysfunction in a rat model [28]. In fact, burn injury dysregulates antioxidant gene expression, leading to declines of total antioxidants, total superoxide dismutase (SOD) activity, and MnSOD activity in rat hearts after burn [26]. Burn-caused mitochondrial damage in mouse heart tissue also correlated with ROS accumulation and increased oxidative stress [29]. This stress can alter the function of proteins, lipids, and DNA. Decreases in myocardial mitochondrial glutathione levels have been observed in severely burned rats alongside increased mitochondrial oxygen consumption and ROS formation [30]. Evidence has also shown that burn injury reduced the activities of antioxidant enzymes MnSOD and glutathione peroxide (GPx), but increased lipid peroxidation in myocardial mitochondria, resulting in enhanced oxidative stress in rat hearts post burn [28,31]. Antioxidant vitamin therapy applied to severely burned rats has been reported to prevent burn-impaired antioxidant defense in myocardial mitochondria, subsequently improving myocardial function [28]. Furthermore, burn injury significantly decreased a subunit of NADH dehydrogenase (a critical component of complex I), associated with decreases in heme oxygenase 1, NADH quinone oxidoreductase 1 glutamatecysteine ligase catalytic subunit, MnSOD, and GpX1 in rat hearts [27]. An emerging study has been performed to identify the specific molecules involved in mitochondrial dysfunction-induced myocardial oxidative injury. One key mitochondrial protein, mitochondrial translation elongation factor Tu (EF-Tumt), has been recognized to contributei to oxidative myocardial damage in rat hearts after severe burns [32]. Downregulation of EF-Tumt in the myocardial mitochondria increased ROS generation and caused myocyte injury [32]. These results confirm that cardiac mitochondrial impairment promotes oxidative stress and cardiac dysfunction after burn trauma.

### 2.3. Burn-Increased Cell Death

**Apoptosis:** In response to burns, myocardial mitochondria begin to swell and lose membrane integrity [28,33], resulting in leakage of cytochrome-c and apoptosis-inducing factor (AIF) from mitochondria into the cytosol. Cytosolic cytochrome *c* and AIF are potent initiators of apoptosis. Increased levels of cytosolic cytochrome-c and AIF with promoted cardiac apoptosis were observed in rodent myocardium after burn [28,33]. Interestingly, stimulating the vagal nerve significantly preserved burn-induced mitochondrial swelling and decreased the release of cytochrome *c* and AIF from mitochondria into the cytosol, thus mitigating mitochondria-activated cardiac apoptosis and reducing myocardial injury after burn [33]. Blockade of the M3-muscarinic acetylcholine receptor or PI3K abolished the vagal nerve stimulation-mediated protection in myocardial mitochondrial response to burn [33]. In addition, mitochondrial ROS (mtROS) and mitochondrial DNA (mtDNA) fragments from damaged mitochondria during stress have been shown to stimulate apoptotic responses in animal studies [34,35,36,37]. Burn injury increases the release of mitochondrial cytochrome *c*, mtROS, and mtDNA into the cytoplasm [38,39,40], inducing cardiac apoptosis. Combating this myocardial apoptosis can be seen through treatments of 17B-estradiol, which preserves mitochondria and decreases cytosolic levels of cytochrome *c*, mtROS, and mtDNA in rat hearts following burn [38]. Furthermore, burn rat serum-induced cardiomyocyte apoptosis is inhibited by the HSP70 protein that interferes with mitochondrial and membrane death receptor pathways [41,42].

**Autophagy:** Autophagy plays a protective role in cellular homeostasis. However, increased autophagy activation occurs and leads to cell death in rat hearts during thermal injury [43]. Autophagic cell death, a non-apoptotic form of programmed cell death [44], has been observed in the myocardium as early as 3 h after severe burn [43]. This study has indicated that the level of an autophagy marker LC3 (microtubule associated protein 1 light chain 3) was significantly changed in the heart at 1-h post-burn and continued to be abnormal through the 12 h post-burn. Importantly, the greatest proportion of cardiomyocyte loss was reported as a derivative from autophagic cell death instead of apoptosis following burn injury [43]. Depression of myocardial function was associated with upregulation of autophagic cell death [43], suggesting that excessive autophagy contributes to myocardia dysfunction. Of note, using rapamycin to increase autophagy has been shown to worsen cardiac dysfunction in burn-injured hearts, while inhibition of autophagy improved burn-depressed cardiac function [43].

**Mitophagy:** Mitophagy is a type of selective organelle autophagy to prevent abnormal mitochondrial accumulation [45]. Receptor-dependent mitophagy is initiated via BCL2/adenovirus E1B 19 kDa protein-interacting protein 3 (BNIP3), FUN14 domain containing 1 (FUNDC1), and NIX (BNIP3-like) [46,47,48], whereas receptor-independent mitophagy is elicited by cytosolic PARKIN translocated into the mitochondria [49]. Under stressful conditions, excessive mitophagy triggers myocyte death and cardiac dysfunction. Following severe thermal injury, the number of impaired mitochondria may increase and exceed the capacity of mitophagy, leading to disrupted mitochondrial hemostasis and resulting in cell death. A recent study has demonstrated that there was no difference in phosphorylated PARKIN, but lower PTEN-induced kinase 1 levels, in swine heart tissue 48 h postburn, suggesting that accelerated mitophagy might occur just before 48 h [22]. However, more studies are required to dissect the role of mitophagy in the pathological process of severe thermal injury and to investigate its temporal nature.

## 3. Molecular Mechanisms Underlying Burn-Impaired Cardiac Mitochondrial Performance

It has been well-recognized that activation of β-adrenergic receptors (AR) via an insistent increase in catecholamine release is responsible for cardiovascular dysfunction after burn injury. In addition, thermal injury initiates alterations in the immune system, leading to excessive production of pro-inflammatory cytokines, including TNF-α, IL-1β, and IL-6. These cytokines along with the activated adrenergic signaling have been shown to impact cardiac mitochondria. Below is an investigative discussion regarding how each react to injury and how they affect mitochondria and cardiovascular function (Figure 2). Although clinical studies have shown that in addition to TNF-α, IL-1β, and IL-6, other common inflammatory mediators such as IL-8 and monocyte chemoattractant protein-1 (MCP-1), etc., are increased within 12–24 h after a burn [9,10], there is no evidence supporting burn injury-induced specific inflammatory factors. In this article, we focus on discussing the influence of TNF-α, IL-1β, and IL-6 on cardiac mitochondria given their important roles in regulating the pathological response to burn [2,4,50].

### 3.1. β-Adrenergic Signaling

Burn injury causes a persistent increase in catecholamine in human patients [51,52,53,54] and animals [2,55]. The elevated levels of catecholamine activate β-ARs, and their activation over prolonged periods can facilitate damage, including cardiovascular dysfunction and mortality after severe burn trauma [2,56]. This activated adrenergic signaling has been shown to induce mitochondrial dysfunction. Impaired cardiac mitochondria can be elicited by β-adrenergic diastolic (Ca^2+^) accumulation, which is mediated via protein kinase A (PKA) activation and subsequently phosphorylates multiple Ca^2+^-cycling proteins [57]. Prolonged β-AR stimulation significantly increased mitochondrial ROS production, resulting in redox modification of the ryanodine receptor and thus increasing the occurrence of arrhythmogenic Ca^2+^ waves in animal experiments [58]. Scavenging ROS during β-AR stimulation normalized sarcoplasmic reticulum Ca^2+^ leak and significantly decreased the generation of Ca^2+^ waves, thereby preventing the occurrence of arrhythmias [58]. Prolonged (24 h) β-adrenergic stimulation also induced mitochondrial membrane depolarization and apoptosis in adult rat cardiomyocytes [59,60]. In addition, acute β-adrenergic stimulation has been reported to increase mitochondrial ROS production in a cAMP- and PKA-dependent but Ca^2+^-independent manner in intact ventricular mouse cardiomyocytes [61]. The mitochondria-targeted antioxidant agent abolished the β-adrenergic inotropic effect, suggesting an important role of this increased mitochondrial ROS in an acute β-adrenergic-induced inotropic response of cardiomyocytes [61].

Notably, the sustained β-AR stimulation leads to increasing nitric oxide (NO) levels after burns [2,62,63]. It has been shown that the NO-cGMP- PKG pathway might be implicated in mitochondria-dependent antioxidant and ischemic preconditioning-induced cardiac protection [64]. Using a PDE5 inhibitor to increase cGMP levels improved cardiac mitochondrial function [21]. Enhancement of cGMP could decrease NADPH oxidase expression/activity, and thus, reduce ROS production [65]. In addition, activation of PKG elicited by ischemic and pharmacological preconditioning conveys cardioprotective signals from the cytosol to mitochondria [66]. Activated PKG initiates the opening of mitochondrial KATP channels, triggering a second PKC pool and causing inhibition of mitochondrial permeability transition and reduction in cell death in animal studies [66,67]. Collectively, these studies suggest that the β-AR-NO-cGMP-PKG pathway can be a possible mechanism by which burn injury induces cardiac mitochondrial dysfunction.

In contrast to increased β-AR activation during the ebb phase (within 48 h post-burn, where cardiac output, metabolism, and oxygen consumption are all decreased) of burn injury [17,18,19,20], decreased β-AR affinity has been observed in rat ventricular muscle 7 and 14 days after burn, associated with a reduced cAMP content [68]. However, another study indicated that there were no significantly changed expression levels of β-ARs (β_1_, β_2_, and β_3_) in the rat left ventricle (LV) 7 days post-burn [69], whereas 7 days post-burn, higher β_1_-AR expression was observed in the burned right ventricle (RV) than in non-burned RV [69] This study implies burn injury-affected β-AR signaling differences between the LV and RV. Given the lack of evidence regarding biventricular differences in β-AR signaling during the ebb phase and longer time periods (>7 days post burn), further studies are required to investigate the influence of burn injury on β-AR signaling and β-AR pathway-altered mitochondrial function in both LV and RV after burn.

### 3.2. TNF-α

Following severe thermal injury, TNF-α is significantly increased in the serum and heart as an acute phase response in burn patients [9,10,70] and animals [50,71,72,73]. Accumulated evidence from our group [74,75,76,77] and others [78,79,80,81,82,83] has demonstrated that locally produced TNF-α is critical for cardiomyocyte impairment and cardiac contractile dysfunction. The pro-inflammatory and pro-apoptotic effects of TNF-α have been well-documented in the heart following injury [74,75,84,85,86,87]. Here, we focus on discussing the impact of TNF-α on cardiac mitochondria from experimental studies. More damaged mitochondria were noticed with focal loss of cristae and disruption of the outer mitochondrial membrane integrity in the mouse heart 12 h after TNF-α challenge [88]. Impaired cardiac mitochondria respiration and decreased complex I activity were also observed in these animals [88]. TNF-α is reported to increase the production of superoxide and hydrogen peroxide (H_2_O_2_) in mitochondria, decrease activities of mitochondrial complexes I, II, and III, and downregulate mitochondrial genes in rat hearts [89]. Mitochondrial dysfunction was further observed with impaired respiration activity, decreased cellular ATP synthesis, and increased mitochondrial superoxide production in a murine cell line of cardiomyocytes (HL-1) exposed to TNF-α [90]. Notably, mitochondrial dynamics (fission and fusion) are related to cardiac dysfunction upon pathophysiological condition. TNF-α has been shown to upregulate dynamin-related peptide 1 (Drp1), thus inducing Drp1-mediated mitochondrial fission and increasing mitochondrial fragmentation in H9c2 cells [91]. Furthermore, TNF-α-promoted mitochondrial ROS production is responsible for TNF-α-induced Ca^2+^ handling disorders in mouse atrial myocytes [92]. Collectively, these studies suggest that TNF-α can impair mitochondrial structure and integrity, damage mitochondria respiration function, reduce ATP synthesis, and increase mitochondria-derived ROS production, thus leading to cardiac dysfunction.

### 3.3. IL-1β

Early experimental studies have demonstrated the synergistic effects of TNF-α and IL-1β on systematic and myocardial damage [79,93,94,95], implying that IL-1β acts as one of “the soluble myocardial depressant factors” in the plasma of septic patients [95,96]. In fact, injection of IL-1β led to decreased LV contractility and relaxation in healthy adult mice, showing a direct link between IL-1β and heart function [97]. The IL-1β-induced negative inotropic effect has also been observed in adult rat ventricular cardiomyocytes with impaired relaxation and is likely mediated by the sphingolipid signaling-regulated calcium transient [98]. Using leptin to modulate the sphingolipid pathway blocks IL-1β-induced cardiosuppression in cardiomyocytes [98]. In addition, IL-1β increases inducible NO synthase (iNOS) activity and thus results in excessive local production of NO, contributing to the abnormality of rat cardiomyocytes [99]. Notably, IL-1β reduces myocardial response to β-AR stimulation [100]. This decreased reaction is mainly attributable to IL-1β-altered calcium channels, particularly the L-type calcium channel [101,102]. This channel is responsible for increasing the calcium current upon β-AR stimulation, but IL-1β uncouples it from the receptor [103]. IL-1β-affected NO production also alters the coupling of the calcium channel with the β-ARs [99,104]. Importantly, IL-1β-stimulated NO production inhibits mitochondrial performance and ATP production in cardiomyocytes, thus decreasing myocardial contractility [105,106,107]. Furthermore, IL-1β also down-regulates the expression of phospholamban, a key regulator of cardiac contractility, reducing sarcoplasmic reticulum calcium-ATPase [108]. Significantly increased IL-1β has been observed in circulation and local myocardium following burn trauma in human patients [9,10,70] and animals [72,109,110]. It is evident that the integrity of myocardial cells is disrupted when exposed to IL-1β as early as one-hour post burn, indicating that IL-1β contributes to myocardial damage following burn injury (96) and possibly to burn-injured cardiac mitochondria.

### 3.4. IL-6

Significantly increased IL-6 levels have been observed in burn patients [111,112,113] and animals (both serum and heart tissue) [109,114,115,116]. Although infusion of IL-6 itself did not alter cardiac contraction and relaxation in Langendorff-perfused rat hearts [109], IL-6 and IL-1β acted synergistically with TNF-α to impair cardiac contractile performance [109]. This combined inflammatory cytokine challenge might simulate what was expected in the heart after burn trauma. IL-6 infusion in rats for 7 days in vivo resulted in a smaller and stiffer LV with increases in cardiomyocyte width and length and LV mass, all of which were most likely due to IL-6-induced myocardial fibrosis [117]. Notably, high IL-6 levels are sustained in burn animals (at 14 days post-burn) and associated with significantly increased heart mass [118], implying that chronic cardiac alterations may occur in burn patients due to sustained high IL-6 levels. On the other hand, accumulated evidence has suggested a key role of IL-6 in metabolic regulation [119]. IL-6 has been reported to regulate peroxisome proliferator-activated receptor-gamma coactivator (PGC)-1α, thus impacting mitochondria biogenesis [119]. Lower cardiac PGC-1α levels were observed in IL-6 knockout mice than in wild type mice fed a high-fat diet [120], and ablation of IL-6 impaired mitochondria synthesis [121]. In addition, IL-6 deficiency decreased mitochondrial oxidative phosphorylation, resulting in cardiac lipotoxin accumulation when the β-oxidation rate of fatty acid remained unchanged [122]. Of note, burn rat serum induced muscle cell death and increased mitochondrial fission and functional impairment, whereas IL-6 antibody treatment reversed these detrimental changes, suggesting a role of IL-6 in impacting muscle cell mitochondria after burn injury [123].

IL-6 also changes with age, increasing steadily after age 50 [124,125]. Although this age-associated increase in IL-6 induced the expression of pro-inflammatory cytokines, there was no cardiac dysfunction noticed in 14-month-old mice [126]. IL-6 deficiency preserved cardiac function in 2-month-old mice but was detrimental in 14-month-old mice after burns complicated by sepsis [126]. Further investigations are needed to study the interaction of IL-6 and cardiac mitochondrial function with aging post-burn and/or complicated by sepsis.

### 3.5. Cardiac Alterations in Calcium Signaling

Normal heart contraction is reliant on a process of excitation–contraction coupling controlled by cytosolic calcium and the power provided by the mitochondria. Mitochondria can impact cardiac contractile activity through the use of calcium in two ways [127]. One is the consequence of mitochondrial calcium uptake and buffering that directly regulates cytosolic calcium transient [127]. The other way is mitochondrial calcium uptake-affected cellular metabolism and energy supplies through the calcium-dependent key enzymes of the tricarboxylic acid cycle and calcium-dependent multiple sites of the electron transport chain, as well as the mitochondrial F_1_/F_0_ ATP synthase [128,129,130]. Calcium transient also plays an important role in regulating intracellular cytoplasmic calcium signals and various cell death pathways [131]. Cytochrome c release from cardiac mitochondria is elicited by increased cytosolic calcium. Burn injury leads to altered calcium handling in animals [132,133]. Hyperactive ryanodine receptors (RyR) have been observed to increase calcium release from the sarcoplasmic reticulum in burn rat serum-challenged cardiomyocytes [134], resulting in significantly increased cytosolic and mitochondrial calcium [135]. Notably, mitochondrial NOS (mtNOS) levels are increased after severe burn injury [136]. Elevated mitochondrial calcium via cytosolic calcium influx following burns might activate mtNOS and result in endogenous NO production [64,137], thus representing a cytosolic calcium-initiated mitochondrial NO-cGMP-dependent apoptosis in cardiomyocytes [64]. Increased cytosolic calcium also resulted in mitochondrial calcium uptake with ATP depletion, contributing to dysregulation of myocardial mitochondrial Ca^2+^ transport in the later stage after burn [138]. This changed Ca^2+^ flux prior to ventricular contractile dysfunction suggests that calcium dysregulation contributes to depressed myocardial function after burn [139]. Growing evidence has further indicated the importance of mitochondrial Ca^2+^ in arrhythmogenesis [140]. Mitochondrial ROS production induces RyR2 oxidation [141] and/or oxidative activation of Ca^2+^/calmodulin-dependent protein kinase II [142], leading to mitochondrial Ca^2+^ overload that contributes strongly to arrhythmogenesis in cardiac pathology. 

Increased mitochondrial calcium uptake significantly impairs mitochondrial performance and cardiac function after injury. Liang et al. observed Ca^2+^ overload in the myocardial mitochondria at 1 h post-burn that remained elevated at 24 h [143]. Accordingly, mitochondrial Ca^2+^ severe overload impaired activity of F_0_F_1_-ATPase and cytochrome *c* oxidase in myocardial mitochondria after thermal injury, thus damaging mitochondrial respiratory function [143]. Taurine was shown to reduce Ca^2+^ overload in the mitochondria and alleviate mitochondrial ROS generation [144]. Pretreatment with a selective inhibitor of mitochondrial calcium also reduced burn serum-increased mitochondrial calcium and inflammatory cytokine production in cardiomyocytes and favored the prevention of burn serum-related myocardial contractile dysfunction [135]. These studies suggest that mitochondrial calcium can modulate myocardial inflammation, ROS production, and cardiac dysfunction in burn injury.

Furthermore, protein levels of the sarcoplasmic reticulum calcium ATPase 2 and RyR and activity of Ca^2+^/Mg^2+^-ATPase and Na^+^/K^+^-ATPase are significantly reduced in the heart after burn injury [139]. Given these proteins implicated in modulating calcium transient, it is further confirmed that calcium dysregulation contributes to mitochondrial impairment and myocardia injury post-burn.

## 4. Other Biological Factors Related to Cardiac Dysfunction and Mitochondrial Dysregulation after Burn

### 4.1. Aging

Aged hearts are at increased risk for oxidative stress and the inability to overcome “adverse events” [145]. Cardiomyocyte numbers also decrease with age, while increasing in size and fibrosis, along with increased ROS and decreased ATP output [146]. Notably, along with age, more mitochondria are unable to retain normal structure and function, leading to less energy output [145,147]. Defects in mitochondrial respiration are shown in the reduced rate of oxidative phosphorylation in complex III or IV in aged hearts [146]. Aging decreases complex III activity selectively in rat cardiac interfibrillar mitochondria via the change of the cytochrome c binding site [148]. In addition, age-related oxidative damage or depletion in the membrane phospholipids likely decreases the activity of complex IV [146]. Furthermore, a 25% decline in complex VIIa has been observed in aging hearts [146]. Similarly, mtDNA is also more frequently mutated over the course of aging, leading to disruption of mitochondrial membrane potential and structural integrity, as well as energy loss [145]. Moreover, aging increases rat cardiac mitochondrial sensitivity to Ca^2+^-induced mitochondrial permeability transition pore opening [149]. Taken together, aging in the mitochondria leads to severe issues with energy production and poor cardiac function in the elderly, along with less tolerance for stress/injury.

Following trauma-hemorrhage, aged rats (22 months old) demonstrated a significant reduction in LV with altered mitochondrial gene expression and impaired mitochondrial performance when compared to younger ones (6 months old) [150,151]. Importantly, aging patients postburn had a higher mortality and a higher incidence of multi organ failure with a variety of dysfunctional/irregular cellular responses when compared to adult patients [152]. These elderly patients showed early signs of hypo-inflammation/reduced inflammatory response, but a hyper-inflammatory response (significantly increased IL-6, TNF-α, and IL-15) two weeks after burn injury [152], indicating an inverse inflammatory response compared with adult patients. The authors did not investigate mitochondrial function in this study. However, they mentioned that analyzing mitochondrial performance in elderly burn patients is necessary for better treatment and survival. In fact, an animal study has shown that increased mortality and morbidity in elderly mice were likely due to the inability of mitochondria to normalize following burn [153]. Decreased MnSOD in the elderly animals led to reduced activity of complex I and IV, decreased ATP, and increased oxidative stress after burn [153]. Therefore, targeting mitochondria for ameliorating burn injury is of great interest in aging patients.

### 4.2. Sex

Although a more favorable outcome has been observed in women (especially young females) and an increased mortality is associated with the male gender following traumatic injuries [154,155,156,157,158,159], the situation is opposite in burn patients. A clinical retrospective study (over a 7-year period conducted on 1611 patients) indicated that the mortality rate following burn injury was 7.2% for male patients versus 13.4% for females [160]. After adjustment by age, the mortality rate among females was still more than twice that of males [160]. In another clinical study involving 4927 burn patients with an overall death of 5.3%, the mortality rate was approximately two-fold higher in women aged between 30 and 59 years compared to the same age group of male burn victims [161]. In fact, it has been shown that female gender is linked to a more detrimental outcome in the majority of clinical studies following burn trauma [6,162,163,164]. The reasons for adverse outcome in women are likely attributable to sex-specific differences in metabolic and neuroendocrine responses following burn injury [161]. In addition, sex-specific inflammatory response has been documented following burns [160,164]. The animal studies [165,166] have shown that male mice exhibited rapid increases in IL-6 associated with the suppression of cell-mediated immune function within 24 h post-burn. After that, cytokine levels decreased and returned to baseline, leading to relatively normal immune function. In contrast, female mice experienced a delay in IL-6 rising with a delayed suppression of immune function in the postburn period [165]. Of note, 17β-estradiol in males or depletion of estrogen by ovariectomy in female mice has been reported to reverse sex-specific differences in immune function following burns [167]. In addition, a greater extent of increased neutrophil activation was observed in male rats than females after burn injury, and castration reduced this activation [168]. These findings suggest that sex hormones play central roles in the immune response to burns. However, other studies did not find sex differences in burn mortality, neither in young nor in aged patients [169,170,171,172]. These conflicting results could be due to mixed groups of patients and variations in burn severity, sample size, age ranges of patients, and the study time period, all of which resulted in difficulties in making valid comparisons.

With respect to the roles of sex and/or sex hormones in mediating myocardial response to injury, animal and human studies from our group [76,173,174] and others [175] have indicated that female hearts exhibited improved post-ischemic cardiac function with decreased myocardial inflammation compared to male hearts. Additionally, exogenous estrogen supplementation provided cardioprotection in male and ovariectomized female animals following acute myocardial ischemia/reperfusion (I/R) injury [176,177]. Moreover, we found better mitochondrial function in female mouse cardiomyocytes compared to male ones, and 17β-estradiol treatment significantly protected male cardiomyocyte mitochondrial performance following I/R or H_2_O_2_ challenge [178]. Similar to these findings in myocardial I/R, preserved LV was observed in female rat hearts compared to male hearts 24 h post-burn [179]. A lesser extent of burn-increased myocardial TNF-α, IL-1β, and Ca^2+^ accumulation was noticed in proestrus/estrus female rat hearts than in diestrus females or males [179]. In addition, using 17β-estradiol post-burn reduced cardiac pro-inflammatory cytokine levels and decreased rat myocardial cellular death [38]. More importantly, 17β-estradiol protected myocardial mitochondria 24 h after burn injury, as demonstrated by improved mitochondrial function and reduced H_2_O_2_ and oxidation of lipids in mitochondria [38]. These findings suggest that acute use of 17β-estradiol post-injury may provide rapid protection for the heart against burn trauma-induced cardiac damage.

## 5. Conclusions

Cardiac dysfunction after severe burn is associated with poor patient outcomes. Understanding the cardiac mitochondrial response to injury would help in the design of novel treatment strategies to overcome the deleterious effects of burn injury on myocardium. Studies targeting the molecular mechanisms of burn-induced cardiac mitochondrial alterations can also provide useful information for deeply understanding the injury response and for developing better methods to restore heart function. Considering that advanced clinical interventions for patients with severe burn trauma result in longer survival, an understanding of cardiac long-term pathophysiological changes and the molecular mechanisms which are involved is of great importance for improving the life quality in severe burn victims. To date, little attention is paid to the long-term pathological impact of severe burn injury on myocardial alterations. Another factor impacting the health of cardiac myocytes is the plasma level of catecholamines, mediators of the hypermetabolic response, which has been shown to significantly increase in severe burn patients and leads to elevated heart rate and myocardial oxygen requirements up to two years [8,51]. Therefore, it is worth investigating the effect of long-term stimulated catecholamine signaling on pathological changes of the myocardium and cardiac mitochondria for better designing treatment approaches.

## Figures and Tables

**Figure 1 ijms-21-06655-f001:**
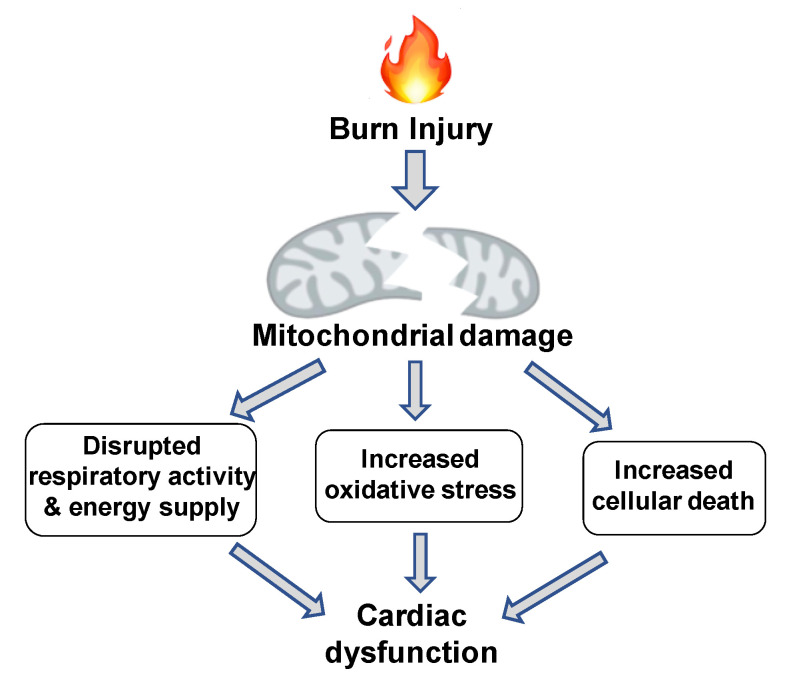
Schematic of burn-induced cardiac mitochondrial pathological alterations.

**Figure 2 ijms-21-06655-f002:**
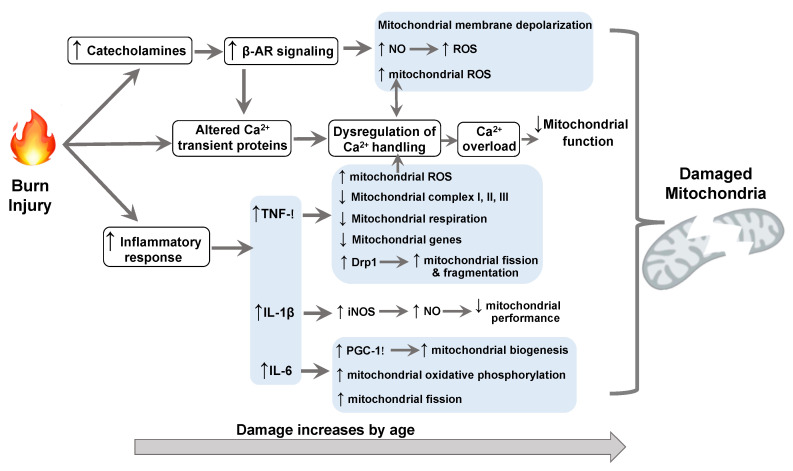
Schematic of potential molecular mechanisms underlying burn-induced cardiac mitochondrial changes. AR: adrenergic receptor; NO: nitric oxide; ROS: reactive oxygen species; Drp1: dynamin-related peptide 1; iNOS: inducible NO synthase; PGC: peroxisome proliferator-activated receptor-gamma coactivator.

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
