# Peer review of "Pathological Responses of Cardiac Mitochondria to Burn Trauma"

_ijms, 2020, doi:10.3390/ijms21186655_

Round 1

Reviewer 1 Report

The review is well structured and the contents are comprehensive.

It would be helpful if the review covers the following issues:

1) What mechanisms lead to the protection of mitochondrial function by sildenafil? I.e. What are the targets in cardiac mitochondria by PKG? Sildenafil was given systemically, are the effects of this PDE5 inhibitor a direct or indirect cause of cardiac phenotype induced by burn injury?

2) Cardiac mitochondria is the prime cause of oxidative stress and apoptosis. How do (acute) decreases in the number, transcription and activity of mitochondria lead to increased oxidative stress? 

3) Following burn injury, are there time-dependent changes in the inflammation and inflammatory cytokines, which are different from those induced by myocardial infarction? TNF-alpha, IL-6 or IL-1beta are common inflammatory cytokines, what are the burn injury-induced specific factors those are more (or as) critical for mitochondrial dysfunction?

4) What type of NOS is involved in cGMP-dependent pathway of myocardium following burn injury?

5) General comments: line 209 " Using a PDE5 inhibitor to lower cGMP levels..."  --> "increase" ; line 280-281, delete "and calcium transient, and
eventually depressing cardiac function". The reference 96 did not perform ca transient or cardiac function experiments.    

Author Response

The review is well structured and the contents are comprehensive.

Response: We sincerely thank you for reviewing our manuscript and for providing insightful suggestions. We have made all revisions along the lines suggested and addressed each below.

It would be helpful if the review covers the following issues:

1) What mechanisms lead to the protection of mitochondrial function by sildenafil? I.e. What are the targets in cardiac mitochondria by PKG? Sildenafil was given systemically, are the effects of this PDE5 inhibitor a direct or indirect cause of cardiac phenotype induced by burn injury?

Response: Thank you for bringing these important questions to our attention.  A recent study has shown that severe burn injury significantly increases myocardial PDE5A expression (both protein and mRNA), decreases cardiac cGMP levels, and reduces PKG protein level and activity. Treatment of sildenafil, an inhibitor of PDE5, normalizes myocardial levels of PDE5A, cGMP and PKG, as well as PKG activity after a burn [1]. Notably, enhancement of cGMP could decrease NADPH oxidase expression/activity, and, thus, reduce ROS production [2].

In addition, activation of PKG elicited by ischemic and pharmacological preconditioning conveys cardioprotective signals from cytosol to mitochondria [3]. Activated PKG initiates the opening of mitochondrial KATP channels, triggering a second PKC pool and causing inhibition of mitochondrial permeability transition and reduction in cell death [3, 4]. However, a detailed mechanism underlying sildenafil-mediated mitochondrial protection following burn requires further investigation.

We have added this information in our revised manuscript (lines 111 – 117 and 236 - 241).

Based on that sildenafil normalizes burn-decreased myocardial PDE5A expression (both protein and mRNA) [1], the effects of this PDE5 inhibitor are likely direct cause to prevent burn-induced cardiac damage in these studies.

2) Cardiac mitochondria is the prime cause of oxidative stress and apoptosis. How do (acute) decreases in the number, transcription and activity of mitochondria lead to increased oxidative stress? 

            Response: The regulation of oxidative stress involves balancing ROS production and ROS scavenging. In addition to inducing cardiac mitochondrial damage, burn injury dysregulates antioxidant gene expression, leading to impairment between oxidant and antioxidant systems. In fact, total antioxidants, total SOD activity and MnSOD activity have been shown to significantly decline in the heart after burn [1]. Therefore, although decreases in the number, transcription and activity of mitochondria, burn injury leads to increased oxidative stress.

This information has been added to line 132 – 134.

3) Following burn injury, are there time-dependent changes in the inflammation and inflammatory cytokines, which are different from those induced by myocardial infarction? TNF-alpha, IL-6 or IL-1beta are common inflammatory cytokines, what are the burn injury-induced specific factors those are more (or as) critical for mitochondrial dysfunction?

Response: There are time-dependent change in the inflammatory cytokine production following burn injury. However, there is no evidence to show burn injury-induced specific inflammatory factors to date. Clinical studies have reported that common inflammatory mediators including TNF-alpha, IL-1beta, IL-6, IL-8, and MCP-1 etc. are increased within 12-24 hours after a burn [5, 6]. Although TNF-alpha, IL-6, and IL-1beta are common inflammatory cytokines in many diseases such as myocardial infarction, they play important roles in regulating pathological response to burn [7-9]. Also, the information is currently unavailable regarding the influence of other common mediators IL-8 and MCP-1 on cardiac mitochondria. In this article, we selected to focus on discussing the effects of TNF-α, IL-1β, and IL-6 on cardiac mitochondria. We have provided a statement about this in the revised manuscript (line 202 – 207).

4) What type of NOS is involved in cGMP-dependent pathway of myocardium following burn injury?

            Response: In this particular paper [10], the authors did not mention the type of NOS in cGMP-dependent pathway of myocardium following burn injury. However, according to another study that mitochondrial NOS (mtNOS) levels are increased after severe burn injury [11] and

increased mitochondrial calcium concentration activates mtNOS, resulting in endogenous NO followed by cGMP production [12, 13], it can be postulated that mtNOS is involved in cGMP-dependent pathway of myocardium following burn injury. We have added this information in the revised manuscript (line 365 – 368).

5) General comments: line 209 " Using a PDE5 inhibitor to lower cGMP levels..."  --> "increase" ; line 280-281, delete "and calcium transient, and eventually depressing cardiac function". The reference 96 did not perform ca transient or cardiac function experiments.    

            Response: We apologize for this omission and thank you for pointing them out. We have changed “lower” to “increase” (line 235) and deleted "and calcium transient, and eventually depressing cardiac function" in the revised manuscript (line 316 – 317).

References:

  1. Wen, J. J.; Cummins, C.; Radhakrishnan, R. S., Sildenafil Recovers Burn-Induced Cardiomyopathy. Cells 2020, 9, (6).
  2. Koupparis, A. J.; Jeremy, J. Y.; Muzaffar, S.; Persad, R.; Shukla, N., Sildenafil inhibits the formation of superoxide and the expression of gp47 NAD[P]H oxidase induced by the thromboxane A2 mimetic, U46619, in corpus cavernosal smooth muscle cells. BJU Int 2005, 96, (3), 423-7.
  3. Costa, A. D.; Garlid, K. D.; West, I. C.; Lincoln, T. M.; Downey, J. M.; Cohen, M. V.; Critz, S. D., Protein kinase G transmits the cardioprotective signal from cytosol to mitochondria. Circulation research 2005, 97, (4), 329-36.
  4. Costa, A. D.; Pierre, S. V.; Cohen, M. V.; Downey, J. M.; Garlid, K. D., cGMP signalling in pre- and post-conditioning: the role of mitochondria. Cardiovasc Res 2008, 77, (2), 344-52.
  5. Jeschke, M. G.; Chinkes, D. L.; Finnerty, C. C.; Kulp, G.; Suman, O. E.; Norbury, W. B.; Branski, L. K.; Gauglitz, G. G.; Mlcak, R. P.; Herndon, D. N., Pathophysiologic response to severe burn injury. Ann Surg 2008, 248, (3), 387-401.
  6. Jeschke, M. G.; Mlcak, R. P.; Finnerty, C. C.; Norbury, W. B.; Gauglitz, G. G.; Kulp, G. A.; Herndon, D. N., Burn size determines the inflammatory and hypermetabolic response. Crit Care 2007, 11, (4), R90.
  7. Carlson, D. L.; Horton, J. W., Cardiac molecular signaling after burn trauma. J Burn Care Res 2006, 27, (5), 669-75.
  8. Auger, C.; Samadi, O.; Jeschke, M. G., The biochemical alterations underlying post-burn hypermetabolism. Biochim Biophys Acta Mol Basis Dis 2017, 1863, (10 Pt B), 2633-2644.
  9. Guillory, A. N.; Clayton, R. P.; Herndon, D. N.; Finnerty, C. C., Cardiovascular Dysfunction Following Burn Injury: What We Have Learned from Rat and Mouse Models. Int J Mol Sci 2016, 17, (1).
  10. Wen, J. J.; Cummins, C. B.; Radhakrishnan, R. S., Burn-Induced Cardiac Mitochondrial Dysfunction via Interruption of the PDE5A-cGMP-PKG Pathway. Int J Mol Sci 2020, 21, (7).
  11. Liang, W. Y.; Tang, L. X.; Yang, Z. C.; Huang, Y. S., Calcium induced the damage of myocardial mitochondrial respiratory function in the early stage after severe burns. Burns : journal of the International Society for Burn Injuries 2002, 28, (2), 143-6.
  12. Dedkova, E. N.; Blatter, L. A., Characteristics and function of cardiac mitochondrial nitric oxide synthase. J Physiol 2009, 587, (Pt 4), 851-72.
  13. Seya, K.; Ono, K.; Fujisawa, S.; Okumura, K.; Motomura, S.; Furukawa, K., Cytosolic Ca2+-induced apoptosis in rat cardiomyocytes via mitochondrial NO-cGMP-protein kinase G pathway. J Pharmacol Exp Ther 2013, 344, (1), 77-84.

Reviewer 2 Report

Dear editor;

In the manuscript, the authors have tried to summarize the pathophysiological response in cardiac mitochondria in burn injury. The topics and theme of manuscript are quite interesting, and authors comprehensively summarize current findings of mitochondrial response in burn injury. However, I have concerned some critical points. Although the manuscript mainly summarizes the findings of basic research regarding burn injury, which seems to be far from actual medical treatments, authors poorly mentioned whether the evidences comes from animal experimental level or human clinical trial level. Thus, because this manuscript would be likely to confuse reader, I would like to recommend to authors to majorly revise the manuscript.

I hope my review would be helpful for your journal.

Thank you,

Masao Saotome, MD, PhD

Assistant professor of cardiology,

Hamamatsu university school of medicine

MAJOR:

  1. First, authors should discuss major causes of death by burn injury. ABA reported that the burn injury death is often caused by burn complications, such as hypovolemic shock, multi-organ failure, respiratory problems, or infection (sepsis). As compared with these complications, at least, the cardiac failure seems to be not major.

  1. In line 97-99, authors should carefully mention whether evidence come from the animal experimental levels or human clinical trials. Because, as mentioned above, the patients with burn injury frequently accompanies hypovolemic shock, it seems to be quite big issue to apply sildenafil.

  1. From line 249-260, in the condition of burn injury, where large amount of multiple chemical stimulants is released, this discussion (protective effects of lower concentration of TNF-alfa) seems to be nonsense.

  1. In the line 326-327, authors first should discuss why the burn injury increase cytosolic and mitochondrial Ca.

  1. In the line 381-382, to discuss gender difference of burn injury, authors should provide the difference in mortality between gender.

MINOR:

  1. In line 36-37, please provide adequate reference.

  1. In line 214, authors should explain what is “ebb phase” of burn injury.

  1. In line 218-219, I do not understand the meaning of the sentence.

  1. In the line 304, authors should mention what is burn serum.

Author Response

In the manuscript, the authors have tried to summarize the pathophysiological response in cardiac mitochondria in burn injury. The topics and theme of manuscript are quite interesting, and authors comprehensively summarize current findings of mitochondrial response in burn injury. However, I have concerned some critical points. Although the manuscript mainly summarizes the findings of basic research regarding burn injury, which seems to be far from actual medical treatments, authors poorly mentioned whether the evidences comes from animal experimental level or human clinical trial level. Thus, because this manuscript would be likely to confuse reader, I would like to recommend to authors to majorly revise the manuscript.

Response: We sincerely thank you for reviewing our manuscript and for providing insightful suggestions. We apologize for our omission not to clearly mention the sources of the evidence (animal experiments or clinical studies) in our manuscript. We have made all revisions along the lines suggested and addressed each below.

MAJOR:

  1. First, authors should discuss major causes of death by burn injury. ABA reported that the burn injury death is often caused by burn complications, such as hypovolemic shock, multi-organ failure, respiratory problems, or infection (sepsis). As compared with these complications, at least, the cardiac failure seems to be not major.

Response: Thank you for the suggestion. We have provided the discussion about major causes of death by burn injury first in the revised manuscript (line 32 -34). However, we disagree with you that “compared with these complications, at least, the cardiac failure seems to be not major”.  The reasons are as follows: 1) impaired cardiovascular function has been noticed as one of the main determinants of acute phase responses of multiple organ dysfunction following severe thermal injury [1]. Autopsy results have shown many pediatric patients present with right heart failure [2]; 2) immediately post-burn patients may have low cardiac values, characteristic of early shock [3]; 3) Clinical studies have demonstrated that poorer outcomes of burn injury are associated with severe cardiac dysfunction [4-6]; 4) the cardiac derangements can culminate in a state of shock based on animal studies [7, 8]; and 5) substantial hemodynamic and cardiac dysfunction following major burn injury contribute to the development of sepsis, multiple organ failure, and death [9]. We have added this information in the 1st and 2nd paragraph of Introduction (line 31 – 34, 41 – 44, and 54 – 57).

  1. In line 97-99, authors should carefully mention whether evidence come from the animal experimental levels or human clinical trials. Because, as mentioned above, the patients with burn injury frequently accompanies hypovolemic shock, it seems to be quite big issue to apply sildenafil.

Response: We are sorry for this omission. We have added this information (lines 108 and 111 - 117). Also, according to review 1’s comment, we have added more information regarding how sildenafil conducts its protective effects in animals following burn injury.

Furthermore, we have clearly mentioned the sources of the evidence (animal experiments or clinical studies) throughout the entire manuscript. (lines 48, 52, 53, 70, 91, 94, 103, 132, 140, 145, 149, 158, 165, 168, 169, 172, 191, 222, 251, 264, 268, 299, 304, 318, 319, 329, 331, 343, 399, 405, 445, 457, 459, 462, 463, and 465).

  1. From line 249-260, in the condition of burn injury, where large amount of multiple chemical stimulants is released, this discussion (protective effects of lower concentration of TNF-alfa) seems to be nonsense.

 Response: Thank you for pointing this out.  We have deleted this discussion (line 285 – 296).

  1. In the line 326-327, authors first should discuss why the burn injury increase cytosolic and mitochondrial Ca.

            Response: Thanks for the suggestion.  We have added the information “Burn injury leads to altered calcium handling in animals [10, 11]. Hyperactive ryanodine receptors have been observed to increase calcium release from the sarcoplasmic reticulum in burn rat serum-challenged cardiomyocytes [12], resulting in  significantly increased cytosolic and mitochondrial calcium [13]” (line 362 – 365).

  1. In the line 381-382, to discuss gender difference of burn injury, authors should provide the difference in mortality between gender. 

Response: We appreciate your suggestion. We have provided this information in the revised manuscript (line 429 – 434).

MINOR:

  1. In line 36-37, please provide adequate reference.

            Response: Thank you. We have added the references (line 38 – 41).

  1. In line 214, authors should explain what is “ebb phase” of burn injury.

            Response: We have added the explanation for “ebb phase” (line 247 – 248).

  1. In line 218-219, I do not understand the meaning of the sentence.

            Response: We have re-organized these sentences to make our point clear (line 252 – 253).

  1. In the line 304, authors should mention what is burn serum.

Response: We have clarified it as burn rat serum (line 342).

References:

  1. Guillory, A. N.; Clayton, R. P.; Herndon, D. N.; Finnerty, C. C., Cardiovascular Dysfunction Following Burn Injury: What We Have Learned from Rat and Mouse Models. Int J Mol Sci 2016, 17, (1).
  2. Pereira, C. T.; Barrow, R. E.; Sterns, A. M.; Hawkins, H. K.; Kimbrough, C. W.; Jeschke, M. G.; Lee, J. O.; Sanford, A. P.; Herndon, D. N., Age-dependent differences in survival after severe burns: a unicentric review of 1,674 patients and 179 autopsies over 15 years. J Am Coll Surg 2006, 202, (3), 536-48.
  3. Cuthbertson, D. P.; Angeles Valero Zanuy, M. A.; Leon Sanz, M. L., Post-shock metabolic response. 1942. Nutr Hosp 2001, 16, (5), 176-82; discussion 175-6.
  4. Williams, F. N.; Herndon, D. N.; Suman, O. E.; Lee, J. O.; Norbury, W. B.; Branski, L. K.; Mlcak, R. P.; Jeschke, M. G., Changes in cardiac physiology after severe burn injury. J Burn Care Res 2011, 32, (2), 269-74.
  5. Jeschke, M. G.; Chinkes, D. L.; Finnerty, C. C.; Kulp, G.; Suman, O. E.; Norbury, W. B.; Branski, L. K.; Gauglitz, G. G.; Mlcak, R. P.; Herndon, D. N., Pathophysiologic response to severe burn injury. Ann Surg 2008, 248, (3), 387-401.
  6. Jeschke, M. G.; Mlcak, R. P.; Finnerty, C. C.; Norbury, W. B.; Gauglitz, G. G.; Kulp, G. A.; Herndon, D. N., Burn size determines the inflammatory and hypermetabolic response. Crit Care 2007, 11, (4), R90.
  7. Horton, J. W.; White, D. J., Diminished cardiac contractile response to burn injury in aged guinea pigs. The Journal of trauma 1993, 34, (3), 429-36.
  8. Wolfe, R. R.; Miller, H. I., Burn shock in untreated and saline-resuscitated guinea pigs. Development of a model. J Surg Res 1976, 21, (4), 269-76.
  9. Abu-Sittah, G. S.; Sarhane, K. A.; Dibo, S. A.; Ibrahim, A., Cardiovascular dysfunction in burns: review of the literature. Ann Burns Fire Disasters 2012, 25, (1), 26-37.
  10. Klein, G. L.; Enkhbaatar, P.; Traber, D. L.; Buja, L. M.; Jonkam, C. C.; Poindexter, B. J.; Bick, R. J., Cardiovascular distribution of the calcium sensing receptor before and after burns. Burns : journal of the International Society for Burn Injuries 2008, 34, (3), 370-5.
  11. White, D. J.; Maass, D. L.; Sanders, B.; Horton, J. W., Cardiomyocyte intracellular calcium and cardiac dysfunction after burn trauma. Crit Care Med 2002, 30, (1), 14-22.
  12. Jiang, X.; Liu, W.; Deng, J.; Lan, L.; Xue, X.; Zhang, C.; Cai, G.; Luo, X.; Liu, J., Polydatin protects cardiac function against burn injury by inhibiting sarcoplasmic reticulum Ca2+ leak by reducing oxidative modification of ryanodine receptors. Free Radic Biol Med 2013, 60, 292-9.
  13. Maass, D. L.; White, J.; Sanders, B.; Horton, J. W., Role of cytosolic vs. mitochondrial Ca2+ accumulation in burn injury-related myocardial inflammation and function. American journal of physiology. Heart and circulatory physiology 2005, 288, (2), H744-51.

Round 2

Reviewer 1 Report

The authors made some changes to the manuscript, which has improved the paper.

I would suggest to include molecular mechanisms of mitochondrial damage in the Figure 2. The current schematic diagram seem superficial.

Author Response

I would suggest to include molecular mechanisms of mitochondrial damage in the Figure 2. The current schematic diagram seems superficial.

Response: We sincerely thank you for reviewing our manuscript and for providing insightful comment again. We have revised figure 2 by including molecular mechanisms of mitochondrial damage based on your suggestion.

Reviewer 2 Report

Authors adequately answered my questions. 

Author Response

Response: We sincerely thank you for reviewing our manuscript again.